# DEEPFAKE CARICATURES: AMPLIFYING ATTENTION TO ARTIFACTS INCREASES DEEPFAKE DETECTION BY HUMANS AND MACHINES

## ABSTRACT

Deepfakes can fuel online misinformation. As deepfakes get harder to recognize with the naked eye, human users become more reliant on deepfake detection models to help them decide whether a video is real or fake. Currently, models yield a prediction for a video's authenticity, but do not integrate a method for alerting a human user. We introduce a framework for amplifying artifacts in deepfake videos to make them more detectable by people. We propose a novel, semi-supervised Artifact Attention module, which is trained on human responses to create attention maps that highlight video artifacts, and magnify them to create a novel visual indicator we call "Deepfake Caricatures". In a user study, we demonstrate that Caricatures greatly increase human detection, across video presentation times and user engagement levels. We also introduce a deepfake detection model that incorporates the Artifact Attention module to increase its accuracy and robustness. Overall, we demonstrate the success of a human-centered approach to designing deepfake mitigation methods.

## 1 INTRODUCTION

Fake or manipulated video ("deepfakes") pose a clear threat in online spaces that rely on video, from social media, to news media, to video conferencing platforms. To the human eye, these computer-generated fake videos are increasingly indistinguishable from genuine videos Nightingale & Farid (2022); Groh et al. (2022). Computer vision models, however, can achieve impressive success at deepfake detection. Here, we explore how best to augment human deepfake detection with AI-assistance.

Currently, AI-assisted deepfake detection relies on using text-based prompts to tell a user that a video is a deepfake. However, recent studies indicate low rates of compliance for these text-based visual indicators: in one study, participants paired with a deepfake detection model updated their response only 24% of the time, and switched their response (from "real" to "fake", or vice versa) only 12% of the time Groh et al. (2022). More innovative approaches have been proposed, such as showing users a heatmap of regions predicted to be manipulated Boyd et al. (2022), but this did not increase acceptance rates relative to text-based indicators. Overall, to make an impact, the development of deepfake detection models must proceed alongside the exploration of innovative and effective ways to alert human users to a video's authenticity.

We present a novel framework that provides strong classical deepfake detection, but crucially also creates a compelling visual indicator for fake videos by amplifying artifacts, making them more detectable to human observers. Because humans tend to be highly sensitive to distortions in faces, we hypothesize that focusing our visual indicator on amplifying artifacts is likely to yield a highly detectable and compelling visual indicator. Our model, "CariNet", identifies key artifacts in deepfakes using a novel *Artifact Attention Module*, which leverages both human supervision and machine supervision to learn what distortions are most relevant to humans. CariNet then generates **Deepfake Caricatures**, distorted versions of deepfakes, using a *Caricature Generation Module* that magnifies unnatural movements in videos, making them more visible to human users.

We make two primary contributions: First, we generate a novel visual indicator for fake videos called Deepfake Caricatures, and show in a user study that they increase human deepfake detection

accuracy by up to 40% compared to non-signalled deepfakes. Second, we develop a framework for identifying video artifacts that are relevant to humans, collect two datasets of human labels highlighting areas that are perceived as fake, and propose a competitive model that leverages this new data.

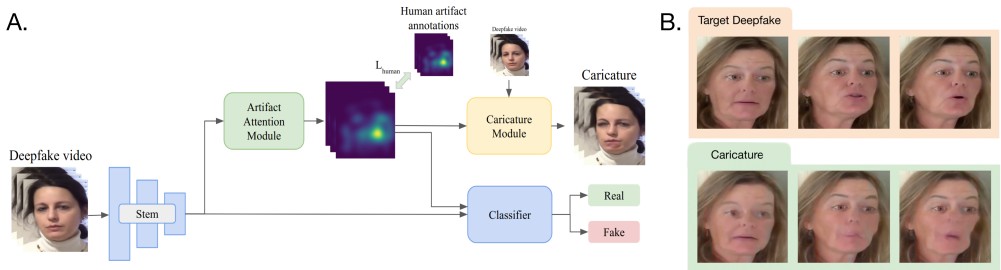

Figure 1: **A.** Overview of our framework. The model learns to identify artifacts visible to humans, then amplify them to generate Deepfake Caricatures: transformations of the original videos where artifacts are more visible. **B.** Example frames of a standard deepfake video (top) and a deepfake caricature (bottom).

## 2 RELATED WORK

**Deepfake detection systems** Deepfake detection is a young but active field. Many novel architectures have been proposed in the last few years to detect videos or images where faces have been digitally manipulated Afchar et al. (2018); Bayar & Stamm (2016); Nguyen et al. (2019b); Sabir et al.; Güera & Delp (2018); Montserrat et al. (2020). Some approaches detect fake faces based on warping or artifacts in the video frames Durall et al. (2019); Li & Lyu (2018); Yang et al. (2019); Li et al. (2020; 2019c). Others detect anomalous biological signals, such as blood volume changes Ciftci & Demir (2019), blinking Li et al. (2018), or eye color and specularity Matern et al. (2019). Recently, models have been augmented with attention mechanisms that highlight specific parts of the input, such as face regions (nose, eyes, etc.) Tolosana et al. (2020), the blending boundary Li et al. (2019c), or regions that are likely to have been manipulated Li et al. (2019a); Stehouwer et al. (2019). Overall, we build on this previous work to develop a novel network that uses attention mechanisms to detect human-relevant artifacts, and amplifies them to increase human detection.

**Human face perception** Humans are exceptionally sensitive to the proportions of faces. Psychology research has shown that faces are encoded in memory based on their deviations from a generic averaged face Benson & Perrett (1991); Sinha et al. (2006), and that the proportions of facial features are at least as important as the particular shape of a facial component for distinguishing among faces Benson & Perrett (1991). This sensitivity is leveraged in the art style known as "caricature", where distinctive features of a face are exaggerated in a way that makes them easier to recognize and remember Benson & Perrett (1991); Mauro & Kubovy (1992); Sinha et al. (2006); Tversky & Baratz (1985), by drawing attention to the facial regions that differ most from the norm. Inspired by these caricatures, our method makes distortions of proportions in fake videos more visible, increasing a viewer's ability to recognize the video as fake.

**AI-assisted decision making** AI decision aids are increasingly being employed for a wide variety of applications Vaccaro & Waldo (2019); Xing et al. (2021); Tschandl et al. (2020). These supplement the judgments of a human user with the output of a machine learning algorithm. Such decision aids improve the accuracy of human decisions, particularly in cases where the AI can detect signals that are complementary to those that humans can detect Tschandl et al. (2020). However, the design of the communication interface between the model and human user is crucial for human acceptance of the model result Tschandl et al. (2020); Deb & Claudio (2015); Hussain et al. (2019); Ancker et al. (2017). We hypothesize that amplifying artifacts in deepfake videos is well-suited for improving human deepfake detection: it targets and amplifies the same information humans would use to make an unassisted judgment, in an easy-to-understand format, without adding irrelevant information.

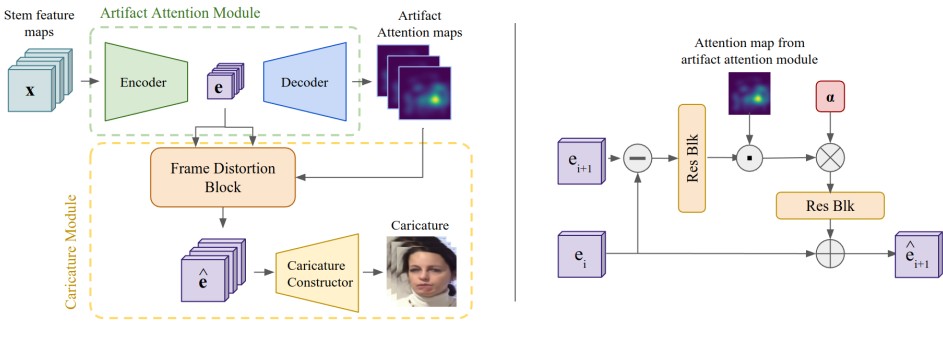

(a) Artifact Attention and Caricature modules      (b) Frame Distortion Block

Figure 2: **Artifact Attention and Caricature Generation modules**. (a) Artifact attention operates with an encoder-decoder architecture to generate artifact heatmaps. These heatmaps are supervised with pre-collected human annotations. The Caricature module receives both the heatmaps and the internal codes $e$, distorts those codes according to the heatmaps, and generates caricatures by reconstructing the video from the distorted codes. (b) The frame distortion block computes the difference between codes $e_i$ and $e_{i+1}$, re-weights it according to the artifact attention maps, and then amplifies it by a factor of $\alpha$ before adding it back to $e_i$ to generate distorted code $\hat{e}_{i+1}$.

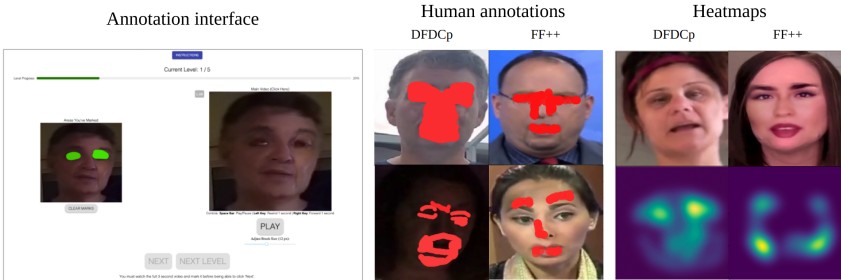

Figure 3: Annotation interface and outputs. Our annotation interface allows users to paint over zones that appear fake. Our system tracks both the position and frame at which an annotation occurs. Users highlighted both large areas and more specific, semantically meaningful areas (e.g., eyebrows).

## 3 MODEL SPECIFICATION

We present a framework that leverages human annotations to detect video artifacts in deepfakes, then amplifies them to create Deepfake Caricatures. Our model, dubbed *CariNet*, uses a combination of self-attention and human-guided attention maps. CariNet contains three main modules (Figure 1)

- An **Artifact Attention Module** that outputs heatmaps indicating the probable location of artifacts in each input frame.
- A **Classifier Module**, which estimates whether the video is fake. This module incorporates the output of the artifact attention module to modulate attention toward artifacts.
- A **Caricature Generation Module**, which uses the Artifact Attention maps to amplify artifacts directly in the videos, yielding *deepfake caricatures*.

### 3.1 ARTIFACT ATTENTION MODULE

This module (Figure 2) guides the model towards regions in the videos that are most likely to contain artifacts. It consists of an encoder-decoder architecture that is partially supervised with human annotations. We incorporate human annotations for two reasons: 1) it biases the model toward the artifacts that are informative to humans, and 2) it guides the module towards regions which may not be locally connected, both of which we hypothesized would yield better caricatures.

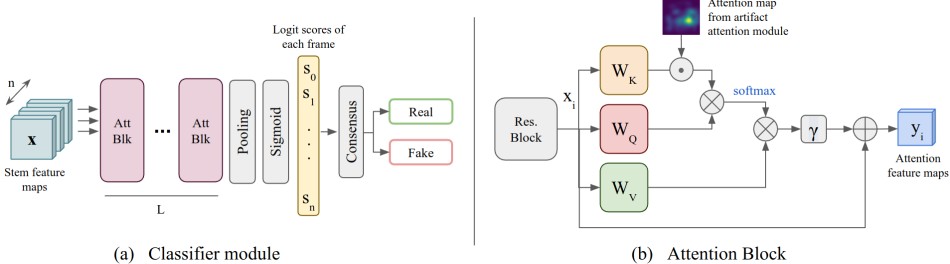

(a) Classifier module          (b) Attention Block

Figure 4: **The Classifier module and its attention blocks**. (a) the classifier takes the feature maps output by the convolutional stem, passes them through attention blocks modulated by human heatmaps, and computes logit scores for each frame before classifying the video. The consensus operation is an average of the logits followed by thresholding to determine the output label. (b) Our attention block: a traditional residual block is followed by key, query and value matrices following the self-attention framework. The key-query product is modulated by our human heatmaps. $\times$ represents matrix multiplication and $\cdot$ represents element-wise multiplication.

**Human-informed ground truth.** We collected human labels on DFDCp and FF++ corresponding to the locations most indicative of doctoring, as perceived by crowd-sourced participants. First, we created a pool of challenging deepfake videos, as these are the most likely to yield non-trivial information about artifacts. From the DFDCp dataset Dolhansky et al. (2019), we selected 500 videos that are challenging for humans (see Supplement), and 500 videos that are challenging for the XceptionNet Rossler et al. (2019) detection model. From FF++, we selected 200 videos for each of the four subsets (Deepfakes, FaceSwap, Face2Face and NeuralTextures). Next, we showed these deepfakes to a new set of participants (mean N=11 per video), who annotated areas of the videos that appeared manipulated (Figure 3). Participants were shown blocks of 100 3-second clips, and used a paint-brush interface to paint on videos regions that appeared unnatural (see Supplement). This resulted in over 11K annotations across 1000 videos for DFDC, and 9k annotations on 800 videos for FF++. For each video clip, we aggregate all annotations and generate one 3D attention map by first applying an anisotropic Gaussian kernel of size (20, 20, 6) in $x$, $y$ and $time$ dimensions, and then normalizing the map of each frame to sum to one. These datasets, including attention maps and individual labels, will be released on our project page upon publication.

**Module details and training**. The artifact attention module is based on the encoder-decoder architectural paradigm, and consists of an Xception-based encoder Chollet (2017) and a 6-block Resnet-based decoder, where upsampling and convolutions are layered between each block (Figure 2). The encoder produces codes $e$ that retain spatial information, and the decoder utilizes those compressed feature maps to generate output heatmaps. We use the human data to supervise the generation of these heatmaps directly. This happens through the sum of three losses: the Pearson Correlation Coefficient, KL-Divergence, and L-1 loss. We confirmed that the Artifact Attention module achieves good performance at reproducing the ground truth maps: predicted maps had an average 0.745 Correlation Coefficient (CC) and a 0.452 KL divergence (KL) with the human-generated maps. In comparison, a simple 2D gaussian achieves 0.423 CC and 0.661 KL.

We trained versions of the Artifact Attention Module with both our DFDCp and FF++ artifact annotations. Qualitatively, the kinds of regions and artifacts that are identified by each Artifact Attention Module are similar. For the modeling results (5), we report the model leveraging the FF++-trained artifact attention module to ensure fair comparisons with previous works. We additionally report the results from the model leveraging the DFDC-trained artifact attention module in the supplement for completeness.

## 3.2 CLASSIFIER MODULE

Our Classifier module (Figure 4) detects if the input is real or fake. This module receives feature maps generated by an EVA-02 Fang et al. (2023) backbone (Stem in Figure 1), a transformer-based model trained to output general visual representations through masked modeling on ImageNet. The stem's embeddings are fed to our classifier module, composed of $L$ Human Attention Blocks fol-

lowed by a global average pooling and a sigmoid activation. We define Attention Blocks as a Residual Block followed by a customized self-attention operation detailed in Figure 4b. We allow this module to supplement its self-attention with the human-guided attention information provided by the Artifact Attention Module, in order to increase attention to key parts of the input (details below). We analyzed Attention block sequence sizes of $L = 18$ and $L = 34$: each alternative yields a different model version, referred to as CariNet-S and CariNet. Cross entropy is used as the loss function. The output of the binary logit function is assigned to each frame as a detection score; we take the averaged score of the whole sequence as the prediction for the video clip.

**Self-attention with artifact heatmaps.** We define our self-attention layers in a similar manner to prior self-attention work Zhang et al. (2018); Bello et al. (2019), but we extended the traditional construction to incorporate modulation from the artifact attention heatmaps. Our self attention layer computes an affinity matrix between keys and queries, where the keys are re-weighted by the artifact attention map, putting more weight on the artifacts that are relevant to humans. Given a feature map $\mathbf{x_i}$ over one frame, and an artifact attention map $A$ over that frame, the module learns to generate an affinity matrix $\mathbf{a_i}$:

$$\mathbf{a_i} = softmax((\mathbf{W_Q x_i})^{\mathbf{T}}(\mathbf{W_K x_i} \odot \mathbf{A})). \tag{1}$$

The softmaxed key-query affinity tensor is then matrix-multiplied with the values $V = W_V x_i$ to generate the output residual $r$. That residual is then scaled by $\gamma$ and added to input $x_i$ to yield the output feature map $y_i$:

$$\mathbf{y_i} = \gamma \mathbf{a_i^T}(\mathbf{W_V x_i}) + \mathbf{x_i}. \tag{2}$$

$\mathbf{W_Q}, \mathbf{W_K}, \mathbf{W_V}$ are learned weight matrices of shape $\mathbf{R}^{\bar{C} \times C}$, $\mathbf{R}^{\bar{C} \times C}$ and $\mathbf{R}^{C \times C}$ respectively, with $\bar{C} = C/4$. $\gamma$ is a learnable scalar which controls the impact of the learned attention feature vis-a-vis the original feature maps $x_i$.

## 3.3 CARICATURE GENERATION MODULE

Finally, the primary contribution of our framework is a novel module for creating Deepfake Caricatures, a visual indicator of video authenticity based on amplification of video artifacts. Figure 1 illustrates the distortion exhibited by our caricatures, but the effect is most compelling when viewed as a video (links to a gallery of caricatures can be found in the Supplement). Our Caricature Generation module leverages the encoder from the artifact attention module, distorts codes with a frame distortion block that operates over every pair of codes $e_i$ and $e_{i+1}$, and uses a Caricature Constructor to create the final caricature (Figure 2). We instantiate the Caricature Constructor as a simple decoder with four blocks composed of 3x3 convolutions followed by nearest neighbor upsampling.

The caricature effect is achieved by amplifying the difference between the representations of consecutive frames, while guiding this amplification with the artifact attention maps generated by the artifact attention module, via element-wise multiplication between the tensor $e_{diff} = ResBlk(e_{i+1} - e_i)$ and the artifact attention map of frame $x_i$ (Figure 2b). This essentially results in a targeted distortion aiming at magnifying the artifacts highlighted by the heatmap.

The distorted code $\hat{e}_i$ of frame $x_i$ is computed as

$$\hat{\mathbf{e}}_{\mathbf{i+1}} = \mathbf{e_i} + \alpha(\mathbf{e_i} - \mathbf{e_{i+1}}) \odot \mathbf{A}, \tag{3}$$

where $\alpha$ is a user-defined distortion factor that controls the strength of the resulting caricature.

## 3.4 LEARNING AND OPTIMIZATION

Training is done in two phases: we first train the classification pipeline (Stem, Artifact Attention Module and Classifier), then freeze the classification pipeline and train the caricature module on a motion magnification task, before combining everything together to generate caricatures.

**Classification pipeline.** Our CariNets were separately trained on the DeepFake Detection Challenge (DFDCp) dataset, FaceForensics++, CelebDFv2 and DeeperForensics. For all datasets, we train on videos from the training set and evaluate on the validation or test set. We randomly sampled

32 frames from videos during training. Fake videos are over-represented in these datasets, so we oversampled real videos during training to achieve real/fake balance. Our CariNets were optimized with Rectified Adam Liu et al. (2019) with the LookAhead optimizer Zhang et al. (2019). We use a batch size of 32 and an initial learning rate of 0.001. Cosine annealing learning rate scheduling was applied with half period of 100 epochs. We chose an early stopping strategy, stopping if validation accuracy stagnates in a 10 epoch window. We apply flipping (probability 0.5) and random cropping (224x224) augmentations. The full loss corresponds to the sum of the loss from the Classifier and Artifact Attention modules (described above).

**Caricature Module.** This module was trained following the motion magnification framework from Oh et al. (2018). We use their synthetic dataset, composed of triplets of frames $(x_i, x_{i+1}, \hat{y}_{i+1})$ constructed to simulate motion magnification. $x_i$ and $x_{i+1}$ correspond to video frames at index $i$ and $i+1$ (respectively), and $\hat{y}_{i+1}$ corresponds to frame $x_{i+1}$ with artificially magnified object displacements. During training, we compute encodings $e_i$ and $e_{i+1}$ with our frozen classification pipeline, and feed each pair to a Frame Distortion Block (2) that learns to generate $\hat{e}_{i+1}$, a distorted version of $e_{i+1}$ where displacements are magnified. The caricature constructor then reconstructs an estimated magnified frame $\bar{y}_{i+1}$, which is compared to the ground truth magnified frame $\hat{y}_{i+1}$ using a L-1 loss (no advantage found for more advanced losses). During this training period with the synthetic dataset, no attention maps are fed to the frame distortion block. After training, when generating a caricature, attention maps are used as a multiplicative modulation affecting the feature maps of the frame distortion block (as shown in Figure 2), which effectively turns magnifications on and off for different parts of the frame. This allows our module to function as a sort of *magnifying glass* over artifacts.

## 4 RESULTS: HUMAN EXPERIMENTS

Here, we introduce Deepfake Caricatures, a novel visual signal that a video is a deepfake based on amplified video artifacts. We demonstrate the effectiveness of Caricatures on user behavior in two ways: first, we compare Caricatures to text-based visual indicators; second, we established the range of conditions where Caricatures successfully boost deepfake detection. See supplement for detailed methods and full statistical reporting.

**Comparison of Caricatures and Text-Based Indicators:** Currently, when news outlets share videos that are known to be deepfakes, they flag the videos using text. However, previous work has shown that users who saw deepfakes flagged using text often continue to believe that the videos were real Boyd et al. (2022); Groh et al. (2022). We tested whether users found Caricatures more convincing than text-based indicators. Fifty challenging deepfakes were selected from the DFDC Dolhansky et al. (2019), along with 50 real videos, and presented to users in a deepfake detection task. Participants (N=30 per condition) were randomly assigned to 1 of 3 conditions: unaided deepfake detection, detection where deepfakes were flagged using text, and detection where deepfakes were flagged using Caricatures. Average hit rates (HR) rates were measures for each condition. Overall, text-based visual indicators improved deepfake detection relative to unaided detection (HR=0.78 and HR=0.53 respectively), although users' performance remained well below ceiling. Crucially, users in the Caricature conditions achieved **hit rates of 0.94**, substantially higher than text-based and unaided detection ($p < 0.0001$ for both). Overall, our results show that vision-based indicators such as Caricatures are much more effective than text-based indicators at changing user behavior.

**Evaluation of Caricatures across conditions:** We next tested whether Caricatures robustly improved performance under a variety of visual conditions (Figure 5). First, we tested how long Caricatures needed to be visible to confer a benefit to users. A random sample of 400 videos from the DFDCp (Dolhansky et al. (2019), 200 real, 200 fake) were presented to participants, for 6 different durations: 300ms, 500ms, 1000ms, 3000ms, and 5000ms. The proportion of times each deepfake video was detected in a sample of 10 participants was calculated. Averaged over all timepoints, deepfake detectability was substantially higher for Caricatures than standard deepfakes, ($F(2, 1630) = 17.12$, $p < 0.001$). Crucially, the advantage for Caricatures over standard deepfakes was significant at every presentation duration ($p < 0.01$ for all). Even with as little as 300 milliseconds of exposure (one third of a second), detection was better for caricatures by 14 percentage points. This advantage increases to 43 percentage points for a 5 second exposure (significant

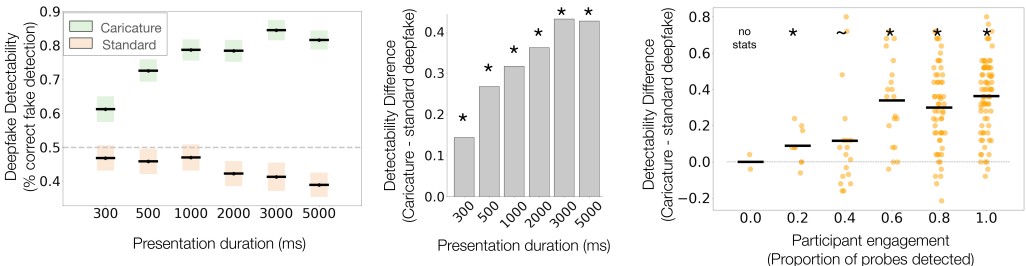

Figure 5: **Behavioral results showing the effect of caricatures on human deepfake detection**. Top Panel: Deepfake detectability by humans for standard (orange) and caricature (green) conditions. Colored boxes represent 95% confidence intervals, and stars indicate that the difference between conditions is significant. Bottom Panel: Improvement in deepfake detection with Caricatures, binned by participant engagement level.

interaction, $F(10, 1630) = 8.88, p < 0.001$). Crucially, while deepfakes were detected no better than chance without visual indicators, Caricatures boosted the likelihood of detection to above 50% across all presentation times. Thus, caricatures increase the likelihood that a deepfake will be detected as fake, even when participants had less than a second of exposure.

We also tested the degree of engagement required for Caricatures to be effective. How closely do users need to pay attention to the videos in order to benefit from this visual indicator? Unbeknownst to participants, *engagement probes* were embedded in the experiment (5 trials per 100-trial HIT), which consisted of standard deepfakes with artifacts which were extremely easy to detect. We reasoned that highly-engaged participants would succeed on all of these trials, but medium to low-engagement participants would miss some proportion of them. Participants were binned based on the proportion of engagement probes they correctly identified as fake, and the detection improvement between caricatures and standard fakes was measured for each bin. We find that Caricatures yield higher detection for 4 out of 5 levels of engagement (no improvement for extremely low engagement levels; 3/4 results significant $p < 0.05$, and 1/4 marginal at $p = 0.051$). Thus Caricatures improve detection even when users are not fully attending to the videos. Overall, these behavioral results demonstrate that the Caricature method is extremely effective at signalling to a human user that a video is fake.

## 5    Results: Model Experiments

While one goal of our framework was to create Deepfake Caricatures, a secondary goal was to assess whether partial supervision from human annotations of artifacts can yield a more performant deepfake detection model. We find that including the human annotations to supplement self-attention during training boosts model performance, leading our models to perform near the state of the art.

### 5.1    Evaluation Details

**Datasets.** We evaluate models on four benchmarks: FaceForensics++ (FF++) Rossler et al. (2019), The Deepfake Detection Challenge Dataset preview (DFDCp) Dolhansky et al. (2019), Celeb-DF v2 Li et al. (2019d), DeeperForensics (DFo) Li et al. (2019d), and FaceShifter (FShifter) Li et al. (2019b). Faces were standardized across datasets, such that each video was $360 \times 360$ pixels, and showed a single face, with a minimum size of 50 pixels and a minimum margin of 100 pixels from the edge of the frame. See Supplement for more details on the datasets and standardization.

**Baseline comparisons.** We evaluate the accuracy of CariNet relative to several established baselines from the literature. For each model, we report performance as published for the datasets it was tested on, and retrain following published model specifications for datasets it was not initially tested on; further details are given in the Supplement.

| Model | CelebDFv2 | DFDCp | FShifter | DFo | Overall |
|---|---|---|---|---|---|
| Xception Rossler et al. (2019) | 73.7 | 65.7 | 72.0 | 84.5 | 75.3 |
| Face X-ray Li et al. (2019c) | 79.5 | 62.1 | 92.8 | 86.8 | 81.2 |
| CNN-GRU Sabir et al. | 69.8 | 63.7 | 80.8 | 74.1 | 73.4 |
| Multi-task Nguyen et al. (2019a) | 75.7 | 63.9 | 66.0 | 77.7 | 71.9 |
| DSP-FWA Li & Lyu (2018) | 69.5 | 64.5 | 65.5 | 50.2 | 63.1 |
| Two-branch Masi et al. (2020) | 73.4 | 64.0 | - | - | - |
| Multi-attention Zhao et al. (2021) | 67.4 | 67.1 | - | - | - |
| LipForensics Zhao et al. (2021) | 82.4 | 70.0 | 97.1 | 97.6 | 86.8 |
| FTCN Zheng et al. (2021) | 86.9 | 74.0 | 98.8 | 98.8 | - |
| DCL Sun et al. (2022) | 82.3 | 76.7 | 92.4 | 97.1 | - |
| RF Haliassos et al. (2022) | 86.9 | 75.9 | 99.7 | 99.3 | - |
| S-B Shiohara & Yamasaki (2022) | **93.2** | 72.4 | - | - | - |
| X+PCC Hua et al. (2023) | 54.9 | 62.7 | - | - | - |
| CariNet (ours) | 88.8 | **76.9** | **99.7** | **99.5** | - |

Table 1: **Detection performance results on unseen datasets.** We report Video-level AUC (%) on four tested benchmarks. All models are pretrained on FF++ (all manipulations). Top results are highlighted in bold, and second-best are underlined. CariNet achieves first or second place on all datasets.

| Method | Train on remaining 3 | | | | |
|---|---|---|---|---|---|
| | DF | FS | F2F | NT | Avg |
| Xception Rossler et al. (2019) | 93.9 | 51.2 | 86.8 | 79.7 | 77.9 |
| CNN-GRU Sabir et al. | 97.6 | 47.6 | 85.8 | 86.6 | 79.4 |
| Face X-ray Li et al. (2019c) | 99.5 | 93.2 | 94.5 | 92.5 | 94.9 |
| LipForensics Haliassos et al. (2021) | 99.7 | 90.1 | 99.7 | 99.1 | 99.5 |
| CariNet (ours) | **99.9** | **99.9** | **99.7** | **99.3** | **99.7** |

Table 2: **Generalization to unseen manipulations.** Video-level AUC (%) on four forgery types of FaceForensics++ (Deepfakes (DF), FaceSwap (FS), Face2Face (F2F) and Neural Textures (NT)).

## 5.2 Deepfake Detection Performance

**Generalization to unseen datasets.** Typically, the performance of deepfake classifiers is assessed primarily based on their ability to generalize well to datasets built with different techniques from what the classifier was trained on Haliassos et al. (2021). Thus, we report our results as our model's ability to train on one dataset and perform well on another. We train models on FF++ and evaluate their performance on CelebDFv2, DFDCp, FShifter and DFo. We note that the DFDCp videos used during evaluation were a different set than the DFDCp videos on which we collected human annotations, to ensure there was no data leak. As is typical Li & Lyu (2018); Haliassos et al. (2021; 2022), we report AUC as it describes model accuracy across a range of decision thresholds. Our CariNets show strong performance in this task, surpassing 13/13 models tested on DFDCp, FShifter and DFo, and 12/13 models on CelebDFv2. We hypothesize that this performance is in part resulting from the attentional framework proposed, which allows CariNet to build a more robust representation of artifact location and properties, focusing as needed on lips, eyes or other telling features. (Table 1).

**Cross-forgery detection** Several methods exist for creating deepfakes, and lead to different artifacts. In order to assess the quality of a detector, we must show good performance across deepfake-generation methods. We divided the FF++ dataset based on the deepfake-generation methods it includes (Deepfake D (2020), FaceSwap Kowalski (2020), Face2Face Thies et al. (2019), and NeuralTextures Thies et al. (2019)), and trained the model on a subset of FF++ containing all four manipulations. We then evaluated its performance on each method independently. In Table 2, we show performance against alternative models from the literature. Overall, our technique is on par with the best-performing model across generation methods.

**Robustness to unseen perturbations** Another key aspect of a forgery detector is the ability to maintain performance even when the input videos have degraded quality, causing new, unseen per-

| Method | Clean | Cont. | Noise | Blur | Pixel |
|---|---|---|---|---|---|
| Xception | 99.8 | 98.6 | 53.8 | 60.2 | 74.2 |
| CNN-GRU | 99.9 | 98.8 | 47.9 | 71.5 | 86.5 |
| Face X-ray | 99.8 | 88.5 | 49.8 | 63.8 | 88.6 |
| LipForensics | 99.9 | 99.6 | 73.8 | 96.1 | 95.6 |
| CariNet | **99.9** | **99.9** | **78.6** | **96.2** | **97.6** |

Table 3: **Generalization performance over unseen perturbations.** Video-level AUC (%) on FF++ over videos perturbed with 5 different modifications. We report averages across severity levels.

| Model | DFDCp | FF++ | CelebDFv2 | DFo | Overall |
|---|---|---|---|---|---|
| CariNet-S w/o att. mechanism | 70.92 | 93.73 | 73.9 | 84.56 | 80.78 |
| CariNet-S w/o modulation from att. module | 72.34 | 94.82 | 76.5 | 91.21 | 83.72 |
| CariNet-S w/ fixed att. (Gaussian) | 68.15 | 90.15 | 71.1 | 82.23 | 77.91 |
| CariNet-S w/ maps from Shiohara (2022) | 71.07 | 93.81 | 74.11 | 84.91 | 80.98 |
| CariNet-S (ours) | **72.90** | **96.81** | **80.1** | **94.33** | **86.04** |

Table 4: **Ablation study results.** We show how certain components of our approach affect video-level AUC (%). We remove the attention mechanism from the Classifier module, we retain the attention mechanism but prevent modulation from the human-informed Artifact Attention module, we replace the human informed modulation with a fixed center bias and replace our heatmaps with a similar approach. All modifications yield lower performance than our proposed network.

turbations. To analyze CariNet's behavior across types of perturbation, we test our model trained with FF++ on test videos from FF++ with 4 perturbations: Contrast, Gaussian Noise, Gaussian blur and Pixelation. We follow the framework of Haliassos et al. (2021) and apply perturbations at 5 severity levels for our comparisons. We report average performance over the 5 severity levels in Table 3. We observe that our method outperforms previous approaches at most severity levels. We hypothesize that our human-guided attention framework might be of help in this setting, as humans are naturally capable of adapting to different lighting conditions, blurs, resolutions and other photometric modifications. This adaptability might be captured in the ground truth maps that guide the learning process of our Artifact Attention module.

**Ablation studies** We confirmed the contribution of adding human supervision via our Artifact Attention module by performing ablation studies across the different datasets under study. We observe performance drops for CariNet-S following ablation in four different scenarios (Table 4): **(1)** removing our custom self-attention blocks (Figure 4) from the Classifier module, and replacing them with simple 3D residual blocks, **(2)** retraining self-attention blocks in the Classifier module, but removing the modulatory input from the Artifact Attention module (this yields regular attention blocks with no key modulation), **(3)** replacing the output of the Artifact Attention module with a fixed center bias, operationalized as a 2D gaussian kernel with mean $\mu = (W/2, H/2)$ and standard deviation $\sigma = (20, 20)$, and **(4)** replacing the output of our attention module with self-blended image masks from Shiohara & Yamasaki (2022) (closest to our approach). Overall, the complete model performed the best, demonstrating the effectiveness of incorporating self-attention modulated by human annotations into deepfake detection frameworks.

## 6    CONCLUSION AND DISCUSSION

This work takes a user-centered approach to deepfake detection models, and proposes a model whose focus is not just to detect video forgeries, but also alert the human user in an intuitive manner. Our CariNet shows excellent detection performance on four different datasets, and crucially, creates novel Deepfake Caricatures which allow for above-chance detection by human observers. Overall, this work establishes the importance of integrating computer vision and human factors solutions for deepfake mitigation. As with any misinformation detection system, there is a risk that our network could be leveraged to produce higher quality deepfakes. However, a system which allows humans to directly detect if a video is doctored will empower them to assess for themselves whether to trust the video. Aggregated over millions of watchers, we believe that the benefits of such a system outweigh the risks.

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
