# OpenReview forum: "Deepfake Caricatures: Amplifying attention to artifacts increases deepfake detection by humans and machines"
_ICLR.cc/2024/Conference — Submitted to ICLR 2024_

### Official Review · Reviewer_iHin · 2023-10-24

**Soundness:** 2 fair
**Presentation:** 2 fair
**Contribution:** 2 fair
**Rating:** 5
**Confidence:** 4

**Summary:**

This paper trains a model on human responses to generate attention maps that amplify artifacts in deepfake videos. They build two datasets of human labels highlighting areas perceived as fake.

**Strengths:**

Overall, the paper is generally well-written, clearly structured, and quite easy to follow. The main strength of the paper is detecting human-relevant artifacts and amplifying them to increase human detection. This task and the technical route are rarely explored.

**Weaknesses:**

1. The words in Fig.1 are too small and hard to read, such as L_{human}.
2. Currently, many deepfake methods are developed to identify fake faces and artifacts region. So, why do we still need to make artifacts more detectable to human observers? It is not clear in this paper.
3. Lots of deepfake methods [1,2,3]  based on attention have been proposed. The author should compare the existing attention mechanisms and the attention used in this paper.

[1] Multi-Attentional Deepfake Detection. CVPR2021
[2] Detection of Deepfake Videos Using Long-Distance Attention. TNNLS2022
[3] An Information Theoretic Approach for Attention-Driven Face Forgery Detection. ECCV2022

**Questions:**

See Weaknesses.

---

> ### Author Response · Authors · 2023-11-22
> **Response to Reviewer iHin**
>
> Thank you for your comments. We respond to your concerns below.
>
> ## **Caption size in Figure 1.**
> We will modify Figure 1 and increase the size of the font corresponding to the loss for the camera ready version.
>
> ## **Need to make artifacts more detectable to human observers.**
> The main objective of our paper is not only to detect fake faces and regions, but to make them _more obvious to human observers_. This helps humans detect fakes early, avoid being fooled and avoid consuming fake information. The core motivation for this, as we describe in the second paragraph of our introduction and in Section 4, is that current methodologies for making a user aware that a video is fake rely on text labels in UIs (e.g. “This Video is Fake”, "Video modified by AI"), and this is suboptimal - easily missed, hardly believed, as many studies show (Introduction, paragraph 2, line 3). Making artifacts more detectable to human observers is key to increasing the chances that humans detect fakes: in fact, our proposed deepfake caricature visual indicator almost _doubles_ human detection performance when compared to the unaided condition, and increases it by 20.5% when compared to a text-based indicator (Section 4, Paragraph 2). This further confirms the necessity for evidencing artifacts to humans instead of just showing text labels. The potential societal impact of this is large: our experiments essentially show that humans choose to not trust a text-based indicator (e.g. "Video modified by AI") **22% of the time**, but only distrust our deepfake caricatures **6% of the time**.
>
> Our paper showcases, through several experiments, that humans are _less easily fooled_ when a technique that amplifies artifacts is used to show them that a video is fake. We hope this clarifies the reason (and the importance) around why we choose to exacerbate artifacts for humans in this work.
>
> ## **Comparison to attention-based deepfake detection methods**
> We already showcase a performance comparison to the attention-based method most aligned with our framework in Table 4 (Shiohara 2022). We additionally provide comparisons to [1] in Table 1, row 7. Further, we bring comparisons with [2,3] in the Table below (cross dataset generalization, FF++ to CelebDF, showing AUC results). We will add these results in the supplemental.
>
> |                   | FF++  | CelebDF |
> |-------------------|-------|---------|
> | Long-Distance [2] | 99.97 | 70.33   |
> | Att-Driven [3]    | 96.94 | 77.35   |
> | CariNet (Ours)    | 99.8  | 88.8    |
>
> In terms of functionality, the proposed attention mechanisms are substantially different from our own. Most rely on an internal measure of self consistency, while ours are sourced from large scale human annotations. Automatically generated heatmaps such as in [1, 2, 3] do not make compelling caricatures, as they highlight face boundaries or the whole swapped area, rather than individual artifacts (unlike our CariNet). Our ablation study shows the effect of simpler synthetic heatmaps (centered gaussian, Table 4) and heatmaps from a relevant work (Shiohara 2022, Table 4). Performance degrades in both cases when our artifact attention module is replaced. We will add short additional insights about attention mechanism differences to our related work section.
>
> We hope these responses helped alleviate your concerns, and we are open to further discussion.

---

### Official Review · Reviewer_7ZWE · 2023-10-25

**Soundness:** 3 good
**Presentation:** 3 good
**Contribution:** 3 good
**Rating:** 6
**Confidence:** 5

**Summary:**

In this paper, the authors propose to make deepfakes seems more 'fake' via the 'Deepfake Caricatures.' Specifically, the authors, relying on manual annotations and heatmaps, guide the model to learn the forged regions within the fake videos. Subsequently, a reconstruction module is applied to introduce additional noise and fluctuations into these regions, aiming to make it clear to viewers that the video is a forgery.

In general, this is an interesting research topic. However, I still have some concerns.

**Strengths:**

1. The proposed 'Deepfake Caricatures' is interesting in Deepfake Detection task.
2. The authors construct an extensive dataset that contributes to both deepfake detection and localization tasks.
3. The experiments validate the effectiveness of the proposed method.

**Weaknesses:**

1. Application and Distinctiveness. The concept of 'Caricatures' in this paper aims to achieve an exaggerated appearance in forged videos. How can this 'Caricatures' concept be distinguished from genuine satirical videos? For instance, if an attacker uses another model trained on 'Caricatures' to introduce disturbances similar to the style in this paper into pristine videos, could it lead viewers to question the authenticity of the pristine videos? In other words, the approach presented in this paper may enhance the distinctiveness of certain fake videos compared to their corresponding real videos in the training dataset. However, does this also result in these fake videos being more challenging to distinguish from some other videos?

2. More qualitative results should be presented in the main manuscript, even though the authors have provided numerous videos in the supplementary materials.

**Questions:**

The primary concern to address is the effectiveness of the method, as discussed in the weaknesses. Does this method enable viewers to effectively distinguish forged videos?

---

> ### Author Response · Authors · 2023-11-22
> **Response to Reviewer 7ZWE**
>
> Thank you for your review and comments. We appreciate the recognition towards our dataset.
>
> We respond to your comments below.
> ### **Application and Distinctiveness.**
> Caricatures are substantially different from satirical videos: they aim at evidencing artifacts instead of generating a satirical depiction of a character. They work by amplifying defects, highlighted by our artifact attention module, across frame embeddings, to generate a magnified version of those defects. They have a particular "uncannyness" in their motion (as can be seen in our gallery) that makes them distinct from a human-crafted satirical video.
> Attackers could potentially generate disturbances similar to the ones exhibited by caricatures to confuse a viewer, but the video itself would be detected as fake by our classifier. Additionally, our caricature framework ensures that only detected fake artifacts are magnified, which reduces the probability that a real video passed through our pipeline results in distortions.
>
> ### **Does our approach make fake videos more challenging to distinguish from other videos?**
> This is an interesting question - our caricatures could be making fake videos more similar to in-the-wild videos through their distortion. We propose that this effect is a) very unlikely to occur, and b) preferable to having fake videos indistinguishable from their real counterparts. First, the distortion applied by our caricatures are very particular: because they stem from a magnification of embedding differences gated by an attentional heatmap, they have a very specific signature that makes them substantially different to other videos in the wild that have been distorted in one way or another. This means that generally, caricatures will not become indistinguishable from another random subset of videos. Second, in the unlikely scenario where they do become hard to distinguish from another set of real distorted videos, we argue that it is still preferable to have fake videos in that state than to disseminate the perfectly crafted fake, where the alteration is made precisely to fool people into thinking they represent real faces (more likely to spread misinformation).
>
>
> ### **Qualitative Results**
> We will ensure that the main manuscript showcases more qualitative examples - thank you for your suggestion. We will add a figure to the appendix presenting more examples. We do highlight, however, that the caricatures are much better appreciated in video form. This is why we decided to use the limited space in the main manuscript to properly showcase our experimental results and figures, leaving most of the visual qualitative exploration to a better suited web gallery.
>
> We hope these comments help address your concerns.

---

### Official Review · Reviewer_Rnom · 2023-10-30

**Soundness:** 2 fair
**Presentation:** 2 fair
**Contribution:** 2 fair
**Rating:** 5
**Confidence:** 5

**Summary:**

The authors propose a face forgery detection model incorporating artificial responses. The aim is to combine machine-learned features with human intuition. The authors use artificial intuition to feel distorted images, similar to an intuition-based data enhancement approach. The whole framework is mainly based on the self-attention mechanism.

**Strengths:**

- The authors are to be commended for building attention maps by introducing human intuition. And the distorted faces are then generated using the generated attention map.
- The authors introduce the generated attention map to the self-attention mechanism for better performance.

**Weaknesses:**

- As the authors state, the primary contribution of the framework is the CARICATURE GENERATION MODULE. However, human intuition can be relatively biased, especially for visually hidden tampering. After all, one of the problems DeepFake detection research is trying to solve is to detect videos that humans can't distinguish.
- The general framework of the method is not NOVEL enough and still uses the usual self-attention mechanism. The focus is on the generation and role of caricatures. And caricatures are more akin to a way of utilizing human abilities to provide auxiliary samples. I'm not sure this comes at a greater cost.
- The experimental results are not solid enough, there are many new methods published in 2023 and the authors should refer to more recent work.

**Questions:**

I noticed a similar paper [1] on arxiv. The two papers adopted the same method and framework, but there were some differences in experimental results. CariNet18 and CariNet34 in [1] should be similar to Carinet-S and CariNet in this paper. The results of CariNet(CariNet34[1]) in the two papers are somewhat different (the performance of this paper is better). However, the results of the ablation study of CariNet-S(CariNet18[1]) (Table 3) were completely consistent.

My question is what changes in the method have brought about the improvement of performance, and why CariNet has been improved while Carinet-S remains unchanged.

[1] https://arxiv.org/abs/2206.00535

---

> ### Author Response · Authors · 2023-11-22
> **Response to Reviewer Rnom**
>
> Thank you for your review. We respond to your main concerns below.
>
> **Bias of human intuition.** We agree that human intuition is biased. We claim, however, that artifact annotations collected by many different humans over many videos _do_ provide strong signal to improve detection and generate effective caricatures. Most experiments in our paper aim at showcasing the usefulness of our artifact attention module and caricature module, both built on human annotations at scale. We repeatedly show the advantage of these human annotations:
> 1. They can improve classification through supervising our artifact attention module, as CariNet beats 13 previous works over 4 datasets and 3 tasks. Our ablation studies in Table 4 further show that using the artifact attention module is better than not using it or replacing it with a different type of attention - crucially, using attention alternatives that are not derived from our human annotations degrades performance.
> 2. They can help humans recognize deepfakes better: Section 4 and Figure 5 show that caricatures, which are a direct product of artifact attention heatmaps, greatly improve human accuracy at the task of recognizing fakes. This improvement is important: humans become 20.5% better at picking up deepfakes with our caricatures (and thus, with the help of our human annotations) than when exposed to text based indicators (e.g. "THIS VIDEO IS FAKE" or "Video Modified by AI")
>
> We recognize that human intuition can be biased, but as we show in this work, using large quantities of human annotations of fake regions improves performance unequivocally, across tasks and across domains. We agree that this result may seem surprising, but we believe that that makes it even more important for the community.
>
> **Novelty.**
> Our framework introduces a larger variety of novelty points than the ones constrained to our attention mechanism. We present the following innovations:
> - a) The integration of artifact attention maps into self-attention structures by directly gating the keys, which to the best of our knowledge has not been previously introduced in any current attention mechanisms (Figure 4B).
> - b) The targeted embedding distortion framework using heatmaps (equation 3) which allows us to amplify specific defects in a video with precision, hijacking the residuals of contiguous time-sliced coded representations built by our encoder (Figure 2).
> - c) Our specific combination of human supervision with machine supervision, leveraging 4 different losses at two key points of the network (to the best of our knowledge, a novel combination of losses), before feeding back the output of one task (artifact attention heatmap reconstruction) as input to the other (classification).
>
> Additionally, our paper introduces **two novel human annotation datasets** (artifacts labeled on DFDCp and FF++) totaling more than 20k annotations on well-known benchmarks, **novel visual indicators** (Caricatures) as well as the framework to make them, and **multiple human experiments** of our own design that showcase the utility of our caricatures.
>
> We hope this helps shed light on the strength of our contributions and the novelty they exhibit, as they go beyond the architectural choice of one part of our backbone.
>
> **Focus on generation and role of caricatures.** We respectfully disagree with the statement that caricatures are closer to "a way of utilizing human abilities to provide auxiliary samples". First, our caricatures are not used as auxiliary samples: the classifier never sees caricatures, and instead leverages the artifact attention heatmaps produced by our artifact attention module (Figure 1A, Figure 4B). Second, our caricatures are much more than auxiliary samples: they are a novel visual indicator that is **20.5% more effective** at evidencing deepfakes to humans than text indicators (Section 4, paragraph 2). That is, when humans are confronted with a fake video, distorting that video with our caricature framework increases hit rates to **0.94** (vs. 0.78 in text based indicator conditions, p<0.0001 for both), making it much more likely that humans will correctly recognize it as fake than showing text that says "THIS VIDEO IS FAKE". We believe this result alone makes caricatures particularly impactful.

---

> > ### Author Response · Authors · 2023-11-22
> > **Response to Reviewer Rnom (cont.)**
> >
> > ### **Strength of experimental results.**
> > The main experimental results of our paper are the measured human hit rates when exposed to caricatures, which convincingly show that our caricatures provoke a substantial increase in human’s abilities to detect fakes (Section 4). These results are contrasted to two typical alternative conditions and our proposal improves on these conditions by a substantial margin (almost doubling their accuracy against unaided detection, and improving against text-based indicators by 20.5%). We additionally showcase detection results, showing that we outperform 13 existing techniques, over 4 datasets, on 3 different tasks (unseen datasets, unseen manipulations, unseen perturbations). We note that 4 of those techniques are quite recent (2022-2023), and that we outperform them. Nevertheless, we showcase additional comparisons to 3 more techniques from 2023 in the table below. Our CariNet outperforms or performs comparably to these techniques on our 4 main datasets. We will add these additional comparisons to the camera ready version.
> >
> > |                | Year | CDFv2 | DFDC | FSh  | DFo  |
> > |----------------|------|-----------|------|------|------|
> > | IID [1]            | 2023 | 83.1      | 75.5 | 98.1 | 98.9 |
> > | AltFreezing [2]    | 2023 | 89.5      | 75.8 | 99.4 | 99.3 |
> > | SFDG [3]           | 2023 | 75.8      | 73.6 | -    | -    |
> > | CariNet (Ours) | 2023 | 88.8      | 76.9 | 99.7 | 99.5 |
> >
> > [1] Huang, Baojin, et al. "Implicit Identity Driven Deepfake Face Swapping Detection." Proceedings of the IEEE/CVF Conference on Computer Vision and Pattern Recognition. 2023. \
> > [2] Wang, Zhendong, et al. "AltFreezing for More General Video Face Forgery Detection." Proceedings of the IEEE/CVF Conference on Computer Vision and Pattern Recognition. 2023. \
> > [3] Wang, Yuan, et al. "Dynamic Graph Learning With Content-Guided Spatial-Frequency Relation Reasoning for Deepfake Detection." Proceedings of the IEEE/CVF Conference on Computer Vision and Pattern Recognition. 2023.
> >
> > We would like to highlight, however, that as we state in our introduction, our goal is not to achieve top automated deepfake detection performance. We aim to build a model that can spatially identify artifacts and generate caricatures, to then test their utility as a forgery indicator; we hope our framework can also be judged by the behavioral impact of our caricatures and the strength of our experimental results in that area.
> >
> >
> > **Questions:**
> > The arxiv mentioned is a draft of our work. This paper constitutes the high quality final version, with heavily updated results, more experiments and improved final models. Once accepted, this paper will replace the arxiv, which will be removed from circulation.
> > - **What changes have brought performance improvements?** We've made changes to our learning and optimization protocol (Section 3.4), upgraded our backbone stem (now using an EVA-02, as mentioned in section 3.2), and collected additional human annotations for FF++ (section 3.1), among others.
> > - **CariNet-S unchanged:** The experiments with CariNet-S are a small scale ablation study, which uses a smaller CariNet without the advanced backbone and with a simpler training protocol. We did not retrain this smaller model for the final version, as the ablation comparisons still hold for this smaller model. As is necessary for this experiment, the same simpler CariNet-S is used for each ablation condition. We will add these details to the camera ready version.

---

### Official Review · Reviewer_qLFh · 2023-11-01

**Soundness:** 3 good
**Presentation:** 4 excellent
**Contribution:** 3 good
**Rating:** 6
**Confidence:** 4

**Summary:**

This paper proposed a method to not only classify deepfake videos but also create caricatures to amplify the artifact area, so that fake video detection can be more easily interpreted by humans.  More specifically, the proposed method consists of three modules: 1) artifact attention module, which is guided by human artifact annotations and to predict artifact region in the video 2) A classifier to classify whether the video is fake or real 3) A caricature module by amplifying the representation difference of consecutive frames around the artifact area, and then applying motion magnification to generate caricatures videos. Experiments are conducted under different scenarios including generalization on unseen datasets, generalization on unseen forgery methods, robustness on unseen perturbations, all showing promising results compared to existing baselines. Moreover, some ablation studies are also provided.

**Strengths:**

--The idea of predicting artifact regions and further amplify artifact regions to create caricatures for deepfake detection are quite interesting.

--The proposed method with the three modules look quite solid. Predicting attention map for artifacts supervised by  human annotated data is somewhat novel to the best of my knowledge.

--The experiments are quite convincing. Promising results on generalization of unseen datasets, forgery methods, and perturbations make me want to try the proposed method.

--The paper is organized quite well and presented clearly. I enjoy reading this paper.

**Weaknesses:**

--Will it help if the classifier takes caricature videos as input, or amplified representations as input?

--Are there any evaluation for the caricatures? E.g., will there be real videos that people will think it is fake after seeing the caricatures? What are percentages when human agree with machines? etc

**Questions:**

See weakness section.

**Details Of Ethics Concerns:**

The paper uses a data set of human artifact annotation.

---

> ### Author Response · Authors · 2023-11-22
> **Response to reviewer qLFh**
>
> Thank you for your comments. We really appreciate your thoughts on the experiments!
>
> Here are our responses to the weaknesses raised:
>
> **Amplified Representations.**
> We tried feeding the caricatures themselves to the classifier in early experiments, but that structure does not improve performance - the caricatures are distorted when they reach the classifier and not helpful enough for that module to be able to effectively utilize it. We did some initial experiments with CariNet-S in earlier stages of the project, and obtained an AUC of **64.5** on DFDCp - compare that to 72.9 when using artifact attention maps. We hypothesize that the signal of artifact location gets mixed with the movement of the pixels themselves. The artifact attention heatmaps, however, are consequential, as shown in our experiments - they are better than attention maps from contemporary papers (Siohara 2022) and synergize well with the modified attention layers of our classification module.
>
> **Caricature Evaluation.** The caricatures are evaluated through our extensive human experiments: we show how humans exposed to caricatures become better at detecting fake videos in Section 4 and Figure 5. We compare to scenarios with no caricatures and with text labels, and observe a significant difference in hit rate when using Caricatures as the visual indicator of fakeness.
>
> Our caricature system will typically not affect a real video, as showcased in our gallery. However, an attacker could hijack our caricature generation module and force a caricature on a real video. In that case, humans could indeed consider that the video is fake. This impression of fakeness could happen in many scenarios however: people will think a video is fake when the lighting is weird, when the people are funny-looking, when the video quality is low, when the subject matter clashes with their beliefs, etc. Given this, to truly answer the question of how often caricatured videos make humans think real videos are fake, we would first need to measure how often this happens without caricatures in the mix, which is out of the scope of this study.

---

### Meta-Review · Area_Chair_zJV7 · 2023-12-06

**Metareview:**

This paper provides a method to detect deepfake videos as well as amplify the artifacts area to make these videos more differentiable by humans. However, some concerns of the reviewers, including the novelty of the framework and the motivation of this submission, are not fully addressed. Half of the reviewers hold negative attitudes toward this submission, therefore, I recommend reject.

**Justification For Why Not Higher Score:**

Concerns are not full addressed.

**Justification For Why Not Lower Score:**

N/A

---

### Decision · Program_Chairs · 2024-01-16

Reject